# Studies of Niobium Sorption from Chloride Solutions with the Use of Anion-Exchange Resins

**Almagul Ultarakova** [ID], **Zaure Karshyga** [ID], **Nina Lokhova, Azamat Yessengaziyev \*** [ID]**, Kaisar Kassymzhanov and Arailym Mukangaliyeva**

The Institute of Metallurgy and Ore Beneficiation, Satbayev University, Almaty 050013, Kazakhstan; a.ultarakova@satbayev.university (A.U.); z.karshyga@satbayev.university (Z.K.); n.lokhova@satbayev.university (N.L.); k.kassymzhanov@satbayev.university (K.K.); 1603sh.arailym@gmail.com (A.M.)
\* Correspondence: a.yessengaziyev@satbayev.university; Tel.: +7-707-7229946

**Abstract:** This paper presents the results of studies for niobium sorption from chloride solutions with the use of anion-exchange organic sorbents: Amberlite IRA-67, Purolite A-100, AB-17-8, and AN-2FN. Niobium sorption was performed from model niobium-containing solutions. Data on comparative sorption characteristics of the studied sorbents were obtained, and the static exchange capacity of the sorbents, values of distribution coefficients, and extraction degree during the niobium sorption from chloride solutions were calculated. The Purolite A-100 anion-exchange resin exhibited the highest affinity for niobium ions under the conditions studied. Its distribution coefficient was 184 mL/g; the niobium extraction degree was 41.5%. To study the equilibrium sorption of niobium from solution on the Purolite A-100 anionite, three well-known models of isotherms were applied: Langmuir, Freundlich, and Dubinin–Radushkevich. The data obtained confirm the good agreement of the Langmuir model with the results of experiments and indicate that the process takes place in a monomolecular layer on the adsorbent having homogeneous adsorption centers. The optimum conditions of niobium sorption by the Purolite A-100 anion-exchange resin were determined as follows: hydrochloric acid concentration—5–10 wt.%, process temperature—35–40 °C, and duration—40–50 min. The calculated activation energy values for niobium sorption from hydrochloric acid solution in the temperature range of 20–50 °C were 25.32 kJ/mol, which corresponds to the intermediate region corresponding to the transition from the diffusion to the kinetic mode.

**Keywords:** niobium; sorption; anion-exchange resin; exchange capacity; distribution coefficient; extraction





## 1. Introduction

Niobium is one of the main components in many heat- and corrosion-resistant alloys. Niobium and its alloys are used as structural materials for parts of jet engines, rockets, gas turbines, chemical equipment, electronic devices, electrical capacitors, and superconducting devices. Niobates of various metals are widely used as segmentelectrics, piezoelectrics, laser materials, and light converters. The main applications of niobium and its alloys are in nuclear power, radioelectronics and chemical apparatus engineering, and vacuum technology. Extraction of such a valuable component as niobium from various raw materials is relevant and timely.

The niobium content in the Earth's crust is $2$–$10^{-3}$% by mass; in nature, it occurs together with tantalum and titanium. Its most important minerals are columbite–tantalite $(Fe,Mn)[(Ta,Nb)O_3]_2$, pyrochlore $(Na,Ca \dots )_2(Nb,Ti)_2O_6[F,OH]$, and loparite $(Na,Ca, Ce \dots )_2(Ti,Nb)_2O_6$.

To open columbite (tantalite) concentrates, decomposition with hydrofluoric acid or alloying with alkalis (NaOH, KOH) is used [1]. Until about 1950, alloying with NaOH was the main way to decompose columbite (tantalite) concentrates. After alloying, the obtained

orthosols of niobium and tantalum were hydrolytically decomposed with water, then the precipitates of polyniobates and polytantalates were decomposed with hydrochloric acid to form metal oxides. Currently, the main method is decomposition with hydrofluoric acid, which results in solutions heavily contaminated with foreign elements, making it difficult to separate pure niobium compounds from them. However, the development of liquid extraction processes with the separation of niobium from associated components has revealed the advantages of acid decomposition, which is characterized by a much shorter process flow. For the processing of loparite concentrates, the widespread method is chlorination, in which ore concentrate is treated with chlorine gas at 750–850 °C in the presence of coal or coke. Low-boiling chlorides of titanium, niobium, and tantalum are captured in condensation units, and then the chloride condensate is either hydrolyzed and purified with further metal separation or pure niobium, and tantalum pentachlorides are obtained with further separation by distillation.

At present, the processing of titanium, tantalum, and tantalum–niobium minerals is accumulating an increasing amount of waste. The recycling of such waste is an urgent task. Unlike conventional niobium raw materials, the niobium content in these wastes is much lower; moreover, the wastes contain a lot of other accompanying macro-components. Processing of such raw materials is expedient using hydrometallurgical processes, such as leaching, sorption, and liquid extraction, which have high selectivity. Hydrometallurgical methods are also more environmentally friendly, do not require a high degree of containment, and are less energy-intensive.

The liquid extraction method is used to extract niobium from technological decomposition solutions of loparite, perovskite, sphene concentrates, and wastes of $LiNbO_3$ production by hydrochloric and sulfuric acids [2]. It should be noted that liquid extraction with organic solvents is an expensive and environmentally harmful method, and in addition to the existing significant advantages, it has disadvantages, the main of which are high toxicity and fire-hazardous organic extractants. All this imposes higher requirements for safety in storage and operation. In addition to this, with the high cost of organic extractants, the question of the possibility of their regeneration is relevant.

Currently, the ion-exchange sorption method is widely used in hydrometallurgy. Its demand can be explained by the fact that it is less expensive, highly selective, highly efficient, and easy to operate and regenerate ion-exchange resin with the provision of environmentally friendly production [3].

However, compared to liquid extraction, niobium sorption is not well studied. There are few studies that show the possibility to extract niobium from solutions by sorption. In studies [4], Nb(V) was sorbed with an inorganic sorbent—pyrolusite. The effect of pH, ionic strength, humic acid, temperature, and time was studied, and it was found that the sorption was affected by the pH of the solution, ionic strength, and humic acid content. Sorption decreased with the increase in ionic strength at acidic pH, and the opposite effect was observed at alkaline pH. The presence of humic acid caused an increase in sorption at acidic pH and a decrease at alkaline pH. Sorption was high at neutral/close to neutral pH (~96%), and lower sorption was observed in acidic (~55%) and alkaline (~85%) media. The high degree of sorption in the pH range of 5–9 was presumably explained by surface complexation between the hydroxyl group of neutral Nb particles, such as $HNbO_3$ or $NbO(OH)_3$, and the surface hydroxyl group (-ON) of pyrolusite.

At most, synthesized organic ion-exchange resins are used for the sorption extraction of niobium from solutions. They are characterized by high strength and chemical resistance. Methods with the use of both anionic and cationic resins for niobium sorption are known.

The proposed method of niobium (V) extraction from fluorinated aqueous solution includes sorption by contact of the solution and the anion-exchange resin. The sorption is performed at pH = 1–4 with AM-2b-grade anion-exchange resin pre-treated with acid or water and containing exchange groups $CH_2$-$N(CH_3)_2$ and $CH_2$-$N(CH_3)_3$ [5]. The studies established that the best results were obtained at pH = 1–4 and acid treatment of the sorbent with sorption time 2 h, SEC = 75–78 mg/g (SEC—static exchange capacity), and pH = 2–3,

and water treatment of the sorbent with sorption time 2 h and SEC = 71–74 mg/g. At the same time, the solution became turbid in 3 h of sorption, and a sediment separated from the solution in a day. The appearance of turbidity in the solution reduced the sorption results. In their other patent [6], sorption was performed at pH = 2–4 by AMP-brand anion-exchange resin. Acid or water treatment of anion-exchange resin was also performed before sorption. The technical result of the invention was to find the optimal conditions for the fast and efficient extraction of niobium from fluorine-containing aqueous solution. The SEC of the resin was 149–160 mg/g at the concentration of the initial solution of 2555 mg/dm$^3$ of niobium.

The results of studies intended to sorb niobium from sulfate–fluoride solutions are presented in [7]. The studies were performed in the concentration range of sulfuric acid 18–54.5% on ANKB-1 and ANKB-7 aminocarboxylic polyampholytes. It was found that the reason for the niobium sorbability decrease with an increase in sulfuric acid concentration was the destruction of strong fluoride anionic complex ions with the formation of neutral and positively charged fluoride and sulfate–fluoride complex ions. The IR-spectroscopic method determined that niobium sorption from mixed sulfate–fluoride solutions took place in the form of fluoride complex anions.

The method of niobium and titanium extraction by sorption from sulfuric acid solution on medium or strong basicity anion-exchange resin (AB-17, AB-16, EDE-10P) is proposed in the patent [8] for the separation of niobium and titanium from iron and the reduction of the corrosive effect of the solution on equipment.

Works [9,10] describe the processing of chloride wastes generated in the titanium slag chlorination process. Niobium sorption under static conditions from the leaching solutions of chloride waste containing up to 2 g/L Nb was considered. The studies included the optimization of process parameters, such as contact time at different niobium concentrations at room temperature. Purolite-C104 and KU-2-8 H cation-exchange sorbents were used to sorb niobium from the solution obtained in the leaching process. When the Purolite-C104 ion-exchange resin was used, the niobium sorption rate from the solution with a concentration of 2 g/L for 3.5 h was about 71.0% (0.071 g/g), and this rate was about 89.0% (0.089 g/g) for the KU-2-8 H ion-exchange resin [9]. The sorption kinetics at three different initial concentrations of niobium (1.0, 1.5, and 2.0 g/L) were studied [10]. The kinetic data obtained matched the pseudo-second-order model ($R^2 \geq 0.989$).

Analysis of available scientific and technical patent literature showed that there are few studies dedicated to niobium sorption with the use of mostly synthetic organic sorbents. Both cation-exchange and anion-exchange resins were used in the studies. As presented above, niobium was sorbed by extraction from acidic fluoride and sulfate–fluoride solutions with the use of anion-exchange resins of different grades and from chloride solutions with the use of cation exchangers. Both positively charged and negatively charged complex niobium ions could be present in chloride solutions, depending on the concentration of hydrochloric acid. According to [11], $[Nb(OH)_2Cl_4]^-$ ions dominated in 6–8 N hydrochloric acid solutions. Positively charged ions were formed as the concentration of $H^+$ ions increased:

$$[Nb(OH)_2Cl_4]^- + H^+ \leftrightarrow [HNbOCl_3]^+ + Cl^- + H_2O \tag{1}$$

Neutral complexes were formed as the concentration of hydrochloric acid decreased:

$$[Nb(OH)_2Cl_4]^- \leftrightarrow Nb(OH)_2Cl_3 + Cl^- \tag{2}$$

A colloidal precipitate was separated due to hydrolysis in solutions with a concentration of less than 3 N HCl:

$$[Nb(OH)_2Cl_4]^- + H_2O \rightarrow Nb(OH)_3Cl_2 + HCl + Cl^- \tag{3}$$

In another paper [12], it is noted that the presence of the compound $[NbOCl_4]^-$ in solutions of 9–10 mol/L HCl was established by the extraction methods.

Articles [9,10] present the results of studies on the niobium sorption from chloride solutions of titanium–magnesium production waste recycling using cation exchangers. The question arises whether it is possible to recycle such solutions using anion exchangers for niobium extraction. As the literature review shows, chloride complex niobium ions at a certain acidity of the solution can also be present in the form of complex anions. According to Equations (1) and (2), depending on the conditions, such as the concentration of the $[Nb(OH)_2Cl_4]^-$ anion, the equilibrium can shift both towards the formation of the $[HNbOCl_3]^+$ cation, the neutral complex $Nb(OH)_2Cl_3$, or the complex anion $[Nb(OH)_2Cl_4]^-$. Presumably, when the $[Nb(OH)_2Cl_4]^-$ anion from the solution interacts with the resin phase and passes to the anionite, the amount of $[Nb(OH)_2Cl_4]^-$ anion in the solution decreases, which should shift the equilibrium in the direction of the initial substance formation, i.e., $[Nb(OH)_2Cl_4]^-$ anion and its further interaction with the ion-exchanger phase. The process, presumably, can be repeated up to a certain final value of the niobium concentration in the solution. It is of interest to study the sorption of niobium from chloride solutions using anion exchangers to what extent the process proceeds.

The main objective of this work was to select an ion exchanger, the study of sorption equilibria, and to find optimal conditions for the effective extraction of niobium anions from hydrochloric acid solutions.

## 2. Materials and Methods

*Materials:* Chemically pure hydrochloric acid; anion-exchange resins made by Amberlite IRA-67, Purolite A-100, AB-17-8, and AN-2FN; and ion-exchange resins made by Purolite, Amberlite, AB-17-8, and AN-2FN according to GOST 20301-74 were supplied to consumers in Cl-form. Table 1 shows the main characteristics of the Amberlite IRA-67, Purolite A-100, AB-17-8, and AN-2FN anion exchangers.

**Table 1.** Characteristics of anion exchangers.

| Brand | Functional Groups | Structure |
|---|---|---|
| Amberlite IRA-67 | Tertiary amino group | Gel |
| Purolite A-100 | Tertiary amino group | Macroporous |
| AB-17-8 | Quaternary trimethylammonium | Gel |
| AN-2FN | Secondary, tertiary aliphatic amino groups, and phenolic groups | Gel |

In the present studies, niobium sorption was performed from a model solution containing 510 mg/L Nb and 10 wt.% HCl.

*Equipment:* IV-6 vibrating grinding mill (LLC "VIBROTECHNIK", St. Petersburg, Russia), reactor equipped with a reflux condenser (Tien-Shyan LLP, Almaty, Kazakhstan), ES VELP stirrer (Velp Scientifica, Usmate, Italy), IKA RW16 stirrer (IKA-WERKE, Staufen im Breisgau, Germany), Shimadzu scales (Shimadzu Corporation, Kyoto, Japan), SNOL drying cabinet (AB "UMEGA", Utena, Lithuania), distiller AE-14-"Ya-FP" (LLC Ferroplast Medical, Yaroslavl, Russia) and LOIP LS-110 stirrer (JSC "LOIP", St. Petersburg, Russia).

*Research methods:* The niobium sorption process was studied under static conditions: a certain amount of ion-exchange resin prepared for the work was placed in a 250 mL conical flask, poured with a solution, and set in a laboratory shaker. Then, stirring was turned on. When the temperature effect on the sorption process was studied, the solution was poured into the conical flask and heated to the required temperature. The ionite was put into the heated solution and stirred.

The most important characteristic of the ion-exchange resin is its exchange capacity that characterizes the amount of substance absorbed by the ion-exchange resin at its saturation [13,14]. Since sorption of niobium from solutions was performed under static conditions, we determined the static exchange capacity (SEC) of the resin, i.e., the capacity of

the resin when equilibrium was reached under static conditions with a solution of a certain volume and composition. Ionite SEC was determined by the following formula, mmol/g:

$$SEC = \frac{(C_0 - C_e) \cdot V}{m} \qquad (4)$$

where $C_0$ is the niobium concentration in the initial solution, mg/L; $C_e$ is the residual equilibrium concentration of niobium in the solution, mg/L; V is the solution volume, L; and m is the mass of dry sorbent, g.

The distribution coefficient $K_d$ expresses the ratio of the parts of the reacting ions in the resin phase and in the solution phase [13]:

$$K_d = \frac{\left[Me^{n+}\right]_{ionite}}{\left[Me^{n+}\right]_{solution}} \qquad (5)$$

where $\left[Me^{n+}\right]_{ionite}$ is the concentration of the sorbed ion in the ion-exchange resin phase, and $\left[Me^{n+}\right]_{solution}$ is the concentration of the sorbed ion in the liquid phase.

The distribution coefficient is the ratio of the number of milligram-equivalents (or millimoles) of the ion sorbed by 1 g of the ion exchanger to the number of milligram-equivalents (or millimoles) of this ion in 1 mL of solution under equilibrium conditions [15,16].

The niobium distribution coefficient during sorption was determined by the formula:

$$K_d = \frac{(C_0 - C_e) \cdot V'}{C_e \cdot m} \qquad (6)$$

where $C_0$ is the niobium concentration in the initial solution, mg/L; $C_e$ is the residual equilibrium concentration of niobium in the solution, mg/L; $V'$ is the volume of solution, mL; and m is the mass of dry sorbent, g.

*Methods of analysis:* Chemical analysis of samples was performed on an optical emission spectrometer with inductively coupled plasma—Optima 8300 DV (Perkin Elmer Inc., Waltham, MA, USA).

## 3. Results and Discussion

*3.1. Comparison of the Sorption Characteristics of Ion-Exchange Resins for the Sorption of Niobium from Acidic Solutions*

The following anion-exchange resins were used: Amberlite IRA 67, Purolite A-100, AB-17-8, and AN-2FN. Data on the comparative sorption capacity of the ion -exchangers obtained as a result of the experiments are shown in Table 2.

**Table 2.** Values of SEC and niobium distribution coefficient of anion-exchange resins of various brands.

| Ionite | Amberlite IRA 67 | Purolite A-100 | AB-17-8 | AN-2FN |
|---|---|---|---|---|
| SEC, mmol/g | 7.43 | 54.81 | 28.80 | 11.15 |
| Distribution coefficient, mL/g | 17 | 184 | 78 | 23 |

From the data in Table 2, it can be seen that the Purolite A-100 and AB-17-8 ion exchangers have a higher sorption capacity than AN-2 FN and IRA. Under equal conditions, sorption is 1.5 times more with the A-100 anion-exchange resin than with AB-17-8, 5.3 times more than with AN-2 FN, and 7.7 times more than with IRA 67. The highest affinity for niobium ions under the studied conditions is exhibited by the Purolite A-100 anion-exchange resin with a distribution coefficient of 184 mL/g. From the data in Table 1, it follows that the Purolite A-100 anionite has a macroporous structure, while the others are of the gel type, which seems to explain its higher niobium capacity compared to the other

studied ion exchangers. Ionites with a macroporous structure are characterized by the presence of macroporous channels with an ordered structure, which may be more favorable for ion exchange in the sorption of large complex anions of niobium $[Nb(OH)_2Cl_4]^-$ on the sorbent. In this case, the diffusion of complex niobium ions inside the resin phase provides better availability of counter ions located throughout the volume of the ion-exchange resin. The sorption can also be influenced by the hydration of ions. At the same time, it is known that anions with a relatively large size are weakly hydrated. The larger the anion and the smaller its charge, it will preferentially pass into the ion-exchange phase [13]. The low hydratability of $[Nb(OH)_2Cl_4]^-$ anion may indicate the possibility of successful use of gel-structured anion exchangers for sorption of the complex niobium anion $[Nb(OH)_2Cl_4]^-$. As can be seen from Table 2, when comparing the other gel-structured anion exchangers, AB-17-8 has the highest capacity, which can be explained by the presence in its structure of quaternary trimethylammonium ionogenic groups, which, compared with secondary and tertiary amines, have a stronger basicity.

The niobium concentration efficiency can be assessed by the extraction degree into the anion-exchange resin (Figure 1). The extraction results on the anion exchangers, all other things being equal, are, as would be expected, in complete agreement with the values shown in Table 2.

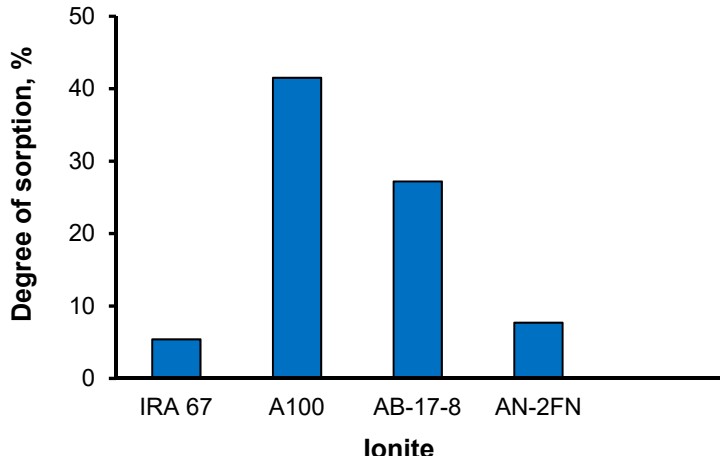

**Figure 1.** Dependence of niobium sorption degree on ion-exchange resin type.

The maximum niobium extraction rate of Purolite A-100 is 41.5%, while that of AB-17-8 is 27.2%, AN-2FN is 7.7%, and Amberlite IRA 67 is 5.4%.

The concentration distribution coefficient of one micro-component is independent of the presence of the other micro-component in solution. According to the Nikolski–Hapon hypothesis [17,18], which has been confirmed by a number of works, the exchange of any pair of ions proceeds independently of the presence of other ions in solution. The micro-component amount sorbed by the ion-exchange ion is directly proportional to its concentration in the solution.

Figure 2 shows the dependence of the SEC of the Purolite A-100 ion exchanger on the concentration of niobium in solution explaining that the higher the ion concentration, the greater the value of the exchange capacity.

It should be noted that the SEC increases insignificantly, only by 14 mg/g, with an increase in niobium concentration from 90 to 400 mg/L.

Thus, the study result of the niobium sorption from hydrochloric acid solutions showed that the sorption is more intense and complete when using the Purolite A-100 anionite with a macroporous structure and the AB-17-8 strong-base gel-type anionite.

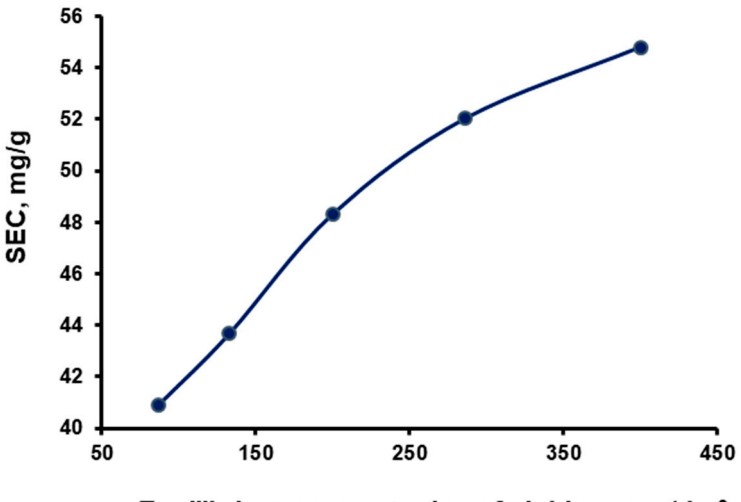

**Figure 2.** Dependence of Purolite A-100 ion-exchange resin on equilibrium concentration of niobium.

*3.2. Study of Sorption Equilibrium*

Studying and developing the sorption process sorption isotherms are of great importance, the use of which can provide information about the mechanism of ion exchange and the nature of the interaction of ions with the resin phase. To describe the adsorption equilibrium, equations based on various ideas about the mechanism of adsorption are now widely used. The mechanism of ion sorption on resin may depend on many factors such as the nature of sorption sites, surface properties, affinity to the sorbent, type of sorbed substance, and others [19,20]. To study the equilibrium sorption of niobium from solution on the Purolite A-100 anionite, this study applied three well-known models: Langmuir, Freundlich, and Dubinin–Radushkevich (D-R).

The theory proposed by the American chemist Langmuir is based on the fact that adsorption occurs in a monomolecular layer, and adsorption is localized on individual adsorption equipotential centers, each of which interacts with only one molecule of adsorbate, with adsorbate molecules that do not interact with each other [21].

The Langmuir isotherm is described by the following equation:

$$q_e = \frac{q_{max} \cdot b \cdot C_e}{1 + b \cdot C_e} \tag{7}$$

The linear form of the Langmuir isotherm equation is as follows:

$$\frac{C_e}{q_e} = \frac{1}{q_{max} \cdot b} + \frac{C_e}{q_{max}} \tag{8}$$

where $C_e$ is the equilibrium concentration of niobium, mg/L; $q_e$ is the amount of niobium adsorbed at equilibrium, mg/g; $q_{max}$ is the adsorption monolayer capacity, i.e., the constant related to maximum absorption capacity, mg/g; and b is the empirical Langmuir constant related to the adsorption energy, L/mg.

Linear dependence represented in coordinates $C_e/q_e$ describing the sorption process of niobium by the Purolite A-100 anionite in accordance with the Langmuir isotherm model is shown in Figure 3.

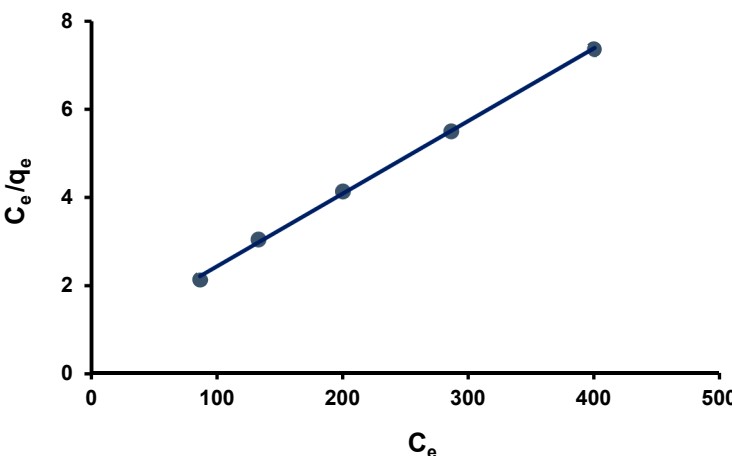

**Figure 3.** Langmuir isotherm for niobium sorption with Purolite A-100 anion exchanger.

Table 3 presents the Langmuir isotherm parameters calculated from the data of the linear $C_e/q_e$—$C_e$ dependence of Figure 3 and Equation (8). As can be seen from Table 3, the calculated maximum capacitance $q_{max}$ is close to the values obtained experimentally at different initial concentrations of niobium in solution (Figure 2). The values of the correlation coefficient $R^2 = 0.999$ testify to good agreement of the Langmuir model with the data obtained experimentally.

**Table 3.** Parameters of isotherms of different models for niobium sorption by Purolite A-100 anionite.

| Isotherm Model | Parameters | Parameter Values |
|---|---|---|
| Langmuir isotherm | Equation | y = 0.0165x + 0.7826 |
| | $q_{max}$ (mg/g) | 60.6 |
| | b (L/mg) | 0.021 |
| | $R_L$ | 0.072 |
| | $R^2$ | 0.999 |
| Freundlich isotherm | Equation | y = 0.1949x + 1.2321 |
| | $K_f$ (L/g) | 17.065 |
| | 1/n | 0.1949 |
| | $R^2$ | 0.9909 |
| Dubinin–Radushkevich isotherm | Equation | y = −0.0026x + 4.467 |
| | $q_D$ (mg/g) | 87.1 |
| | $B_D$ (mol$^2$/kJ$^2$) | 0.0026 |
| | E (kJ/mol) | 13.87 |
| | $R^2$ | 0.9913 |

Another criterion for evaluating the adsorption capacity of a sorbent with respect to a particular sorbate is the dimensionless equilibrium parameter $R_L$, which can be used to predict whether the sorption system will be favorable or unfavorable. This parameter is estimated from the basic characteristics of the Langmuir equation and can be calculated using the following equation [22,23]:

$$R_L = \frac{1}{1 + bC_o} \tag{9}$$

where b is the Langmuir constant, and $C_o$ is the initial concentration of niobium in solution.

The value of $R_L$ indicates the nature of the isotherm: irreversible ($R_L = 0$), favorable ($0 < R_L < 1$), linear ($R_L = 1$). or unfavorable ($R_L > 1$). The obtained value of $R_L$ was 0.072, which indicates that the sorption of niobium from solution by the Purolite A-100 anion-exchange resin, is a sufficiently favorable process, which is described by the Langmuir isotherm model.

By substituting the graphically found value of the maximum capacity qmax and the Langmuir constant b into Equation (7), the adsorption equation, which obeys the Langmuir isotherm model and allows the calculation of the Purolite A-100 anionite niobium capacity at given equilibrium concentrations of niobium in solution, can be determined:

$$q_{calc} = \frac{60.6 \cdot 0.021 \cdot C_e}{1 + 0.021 \cdot C_e} \tag{10}$$

The Freundlich isotherm model is based on the assumption that adsorption occurs on a heterogeneous surface with a non-uniform energy distribution of the active centers, accompanied by interactions between the adsorbed molecules [24]. The Freundlich isotherm is represented by the equation:

$$q_e = K_f \cdot C_e^{\frac{1}{n}} \tag{11}$$

where $C_e$ is the equilibrium concentration of niobium, mg/L; $q_e$ is the amount of niobium adsorbed at equilibrium, mg/g; $K_f$ is the Freundlich constant related to the adsorption capacity, L/g; n is the constant related to the energy of adsorption intensity; and the value $1/n$ gives an indication of the favorable adsorption.

In its logarithmic form, in the coordinates of the linear equation log $q_e$ from log $C_e$ the Freundlich equation would look as follows:

$$\log q_e = \log K_f + \frac{1}{n} \log C_e \tag{12}$$

The results obtained from the linear relationship in the log $q_e$–log $C_e$ coordinates (Figure 4) according to the Freundlich isotherm (12) also show a high correlation with experimental data $R^2 = 0.9909$ (Table 3), slightly inferior to the Langmuir isotherm model ($R^2 = 0.999$). The Freundlich constants $K_f$ and $1/n$ are calculated from the slope and intersection of the log$q_e$ and log $C_e$ plots (Figure 4) and the adsorption parameters are shown in Table 3. A value of $1/n$ less than 1 indicates that niobium is well adsorbed by the Purolite A-100 anionite.

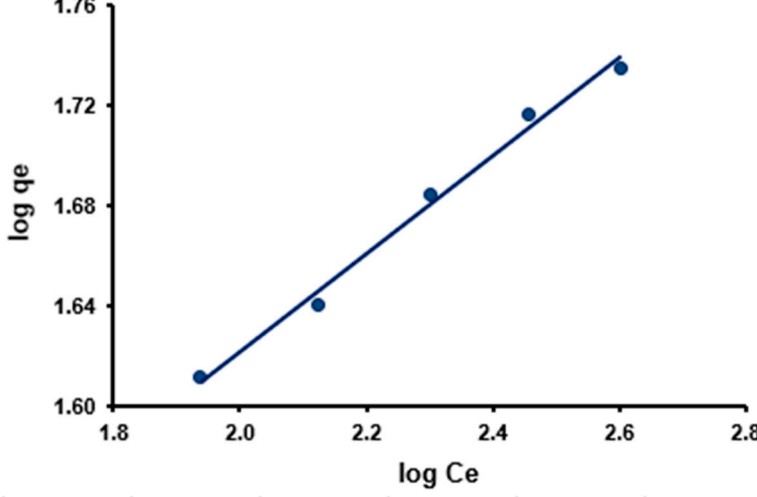

**Figure 4.** Freundlich isotherm for niobium sorption with Purolite A-100 anionite.

The $K_f$ and $1/n$ parameters of Equation (12) found graphically make it possible to find the adsorption equation from which the amount of niobium adsorbed from solution under

equilibrium conditions can be calculated. For niobium sorption with the Purolite A-100 anion-exchange resin, the equation would look as follows:

$$q_{calc} = 17.065 \cdot C_e^{0.1949} \tag{13}$$

The Dubinin–Radushkevich isotherm model assumes a Gaussian energy distribution over a heterogeneous surface, which helps to establish the physical or chemical nature of adsorption [25]. It is more general with respect to the Langmuir model as it does not assume surface homogeneity and constancy of the adsorption potential [26,27]. The Dubinin–Radushkevich isotherm equation is expressed in the following form [28,29]:

$$\ln q_e = \ln q_{DR} - B_{DR}(\varepsilon)^2 \tag{14}$$

$$\varepsilon = RT \ \ln\left(1 + \frac{1}{C_e}\right) \tag{15}$$

where $q_e$ (mg/g) is the amount of adsorbed niobium ions at equilibrium concentration; $q_{DR}$ (mg/g) is the theoretical saturation capacity; $B_{DR}$ (mol$^2$/kJ$^2$) is the Dubinin–Radushkevich isotherm constant related to adsorption energy; $\varepsilon$ is the Polanyi potential (kJ/mol), reflecting the isothermal work of transfer of one mole of niobium from the volume of equilibrium solution to sorbent surface; $C_e$ (mol/L) is the concentration of niobium in solution at equilibrium; R ($8.314 \times 10^{-3}$ kJ/mol·K) is the universal gas constant; and T (K) is the absolute temperature.

The linear dependence obtained in coordinates $\ln q_e$-$\varepsilon^2$, describing the process of niobium sorption from solution on the Purolite A-100 anionite in accordance with the model of isotherm D-R, is presented in Figure 5. The value of the constant $q_{DR}$ is calculated from the point of intersection of the linear relationship (Figure 5) with the ordinate axis ($\ln q_e$) and the value of the constant $B_{DR}$ from the tangent of the slope of the linear relationship with the abscissa axis ($\varepsilon^2$).

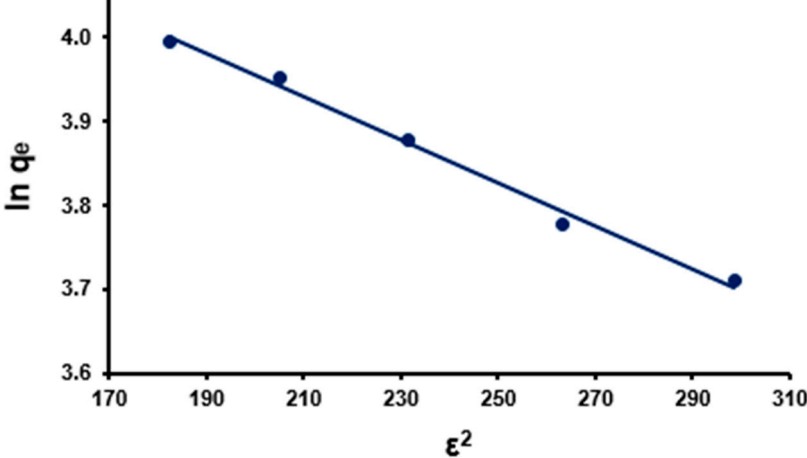

**Figure 5.** Dubinin–Radushkevich isotherm for niobium sorption with anionite Purolite A-100.

The Dubinin–Radushkevich model indicates the nature of the adsorption of the adsorbate on the adsorbent and can be used to calculate the average free energy of sorption. The Dubinin–Radushkevich $B_{DR}$ isotherm constant gives an indication of the average sorption free energy E (J/mol) per mole of niobium as it moves onto the sorbent surface from an infinite distance in solution. E can be determined from the equation [30]:

$$E = \frac{1}{\sqrt{2 \cdot B_{DR}}} \tag{16}$$

If the E value is between 8 and 16 kJ/mol, the sorption process proceeds by the ion-exchange mechanism; if the E value is less than 8 kJ/mol, the sorption process is of physical nature. The obtained value of the free energy amounted to 13.87 kJ/mo1, indicating the chemical nature of the interaction when sorption of niobium on the Purolite A-100 anionite. The value of the correlation coefficient $R^2$ was 0.9913. However, the theoretical saturation capacity $q_D$ was 87.1 mg/g, which is much higher than the experimental data. As it is known, the Dubinin–Radushkevich isotherm model is designed mainly to study adsorption on microporous surfaces. As can be seen from the data in Table 1, Purolite A-100 has a macroporous structure.

The sorption of niobium on the Purolite A-100 anionite also showed good comparability with the Freundlich isotherm model, the correlation coefficient $R^2$ was 0.9909. At the same time, it should be taken into account when selecting the most appropriate model of the sorption process that the Freundlich isotherm is based on the fact that sorption occurs on a heterogeneous surface with an uneven energy distribution of the active centers. In the case of the Purolite A-100 anionite having only a tertiary amine functional group active (Table 1), it can be assumed that the active sites, which in this case are tertiary amine functional groups, are energetically homogeneous and sorption of niobium occurs on equipotential sorbent centers.

Results of the studies indicate that the most appropriate isotherm model is Langmuir, characterized by the highest correlation coefficient $R^2 = 0.999$ (Table 3), showing good agreement with experimental data and indicating that the process occurs in the monomolecular layer on the adsorbent, which has homogeneous adsorption centers.

### 3.3. Determination of Optimal Niobium Sorption Conditions

The anion exchangers under consideration contain a salt of a quaternary ammonium base as an ionogenic group, as well as secondary and tertiary amino groups. In an acid medium, the amino groups attach a proton [31], then form an alkylammonium salt with the anion from the solution. The reactions of interaction of anion exchangers with the complex anion of niobium $[Nb(OH)2Cl4]^-$ can be represented as follows:

- For anion exchangers that contain tertiary amine and secondary amine functional groups:

  (a)  Tertiary amine reactions (Purolite A-100, Amberlite IRA 96, and AH-2FN)

$$RNR'R'' + H^+ = RNR'R''H^+ \tag{17}$$

$$RNR'R''H^+ + [Nb(OH)_2Cl_4]^- = RNR'R''H[Nb(OH)_2Cl_4] \tag{18}$$

  (b)  Secondary amine reactions (AN-2FN)

$$RNR'H + H^+ = RNR'H_2^+ \tag{19}$$

$$RNR'H_2^+ + [Nb(OH)_2Cl_4]^- = RNR'H_2[Nb(OH)_2Cl_4] \tag{20}$$

- For AB-17-8 anionite containing the quaternary trimethylammonium functional group:

$$RN(CH_3)_3Cl + [Nb(OH)_2Cl_4]^- = RN(CH_3)_3[Nb(OH)_2Cl_4] + Cl^- \tag{21}$$

where R, R′, and R″ are hydrocarbon radicals (or framework).

As can be seen from Reactions (17) to (20), the acidity of the medium is very important for ion exchange for secondary and tertiary ionogenic groups, which work only in an acidic medium. In this connection, it was of interest to study the acid concentration.

The effect of hydrochloric acid concentration on the niobium sorption process was studied in the acid concentration range of 5–20 wt.% at 35 °C with the process duration of 50 min. The experimental results are shown in Figure 6.

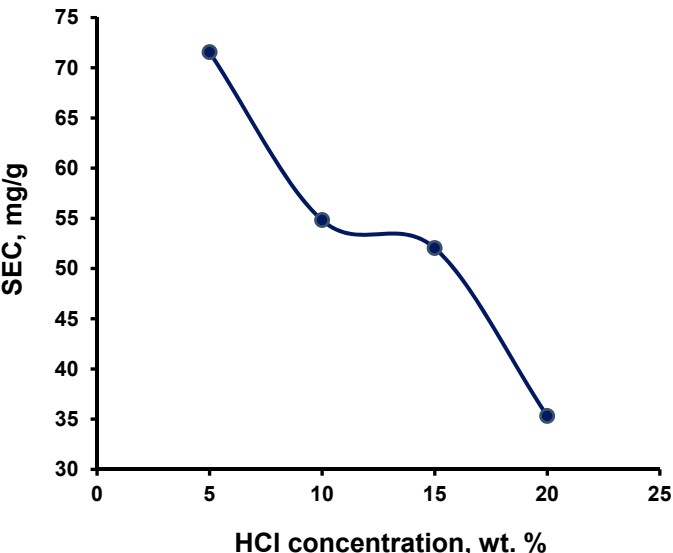

**Figure 6.** Dependence of Purolite A-100 ion-exchange resin on hydrochloric acid concentration.

The exchange capacity of the Purolite A-100 anion-exchange resin changes significantly with an increase in hydrochloric acid concentration in the solution.

The SEC value is 71.5 mg/g at a concentration of 5 wt.%, and the SEC value varies slightly at ~52–55 mg/g in the range of 10–15%. A further increase in the hydrochloric acid concentration results in a decrease in the SEC to 35.3 mg/g, which apparently can be attributed to the formation of positively charged complex niobium ions $[HNbOCl_3]^+$ Rreaction (1) [11].

The process was not carried out with concentrations of HCl below 5 wt.% to avoid a colloidal precipitate due to hydrolysis of the complex anion $[Nb(OH)_2Cl_4]^-$ in Reaction (3) [11].

The SEC values at hydrochloric acid concentrations from 5 to 10 wt.% can be considered satisfactory, and there is no need to neutralize the acid, which can be returned to the preparation of a niobium-containing solution.

The kinetic properties of the studied ion exchangers were determined. Figures 7–10 show the dependences of the SEC on the duration of contact of the ion-exchange resin with the solution.

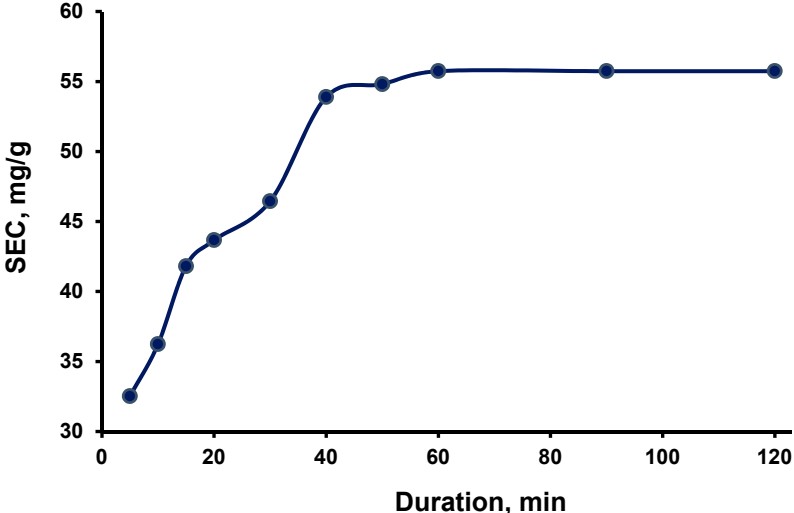

**Figure 7.** Dependence of SEC value on the duration of contact on Purolite A-100 anion-exchange resin.

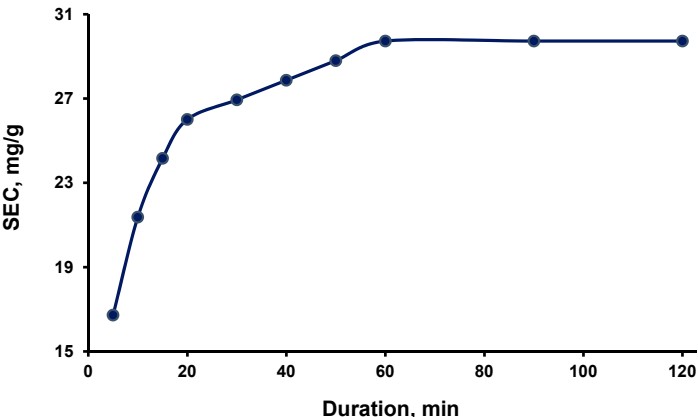

**Figure 8.** Dependence of SEC value on contact duration on AB-17-8 anion-exchange resin.

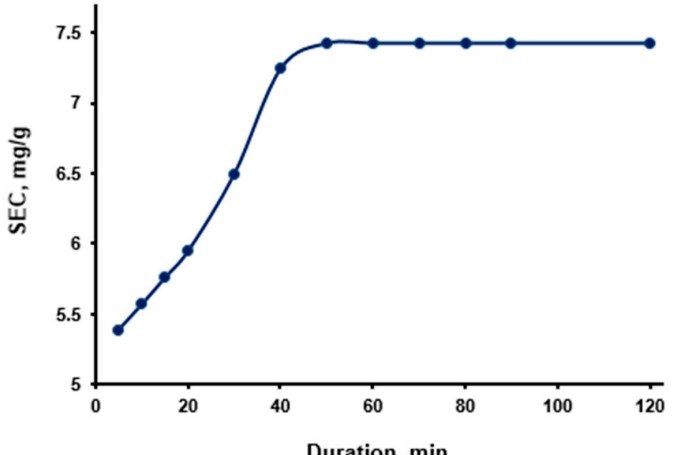

**Figure 9.** Dependence of SEC value on contact time with Amberlite IRA 67 anion-exchange resin.

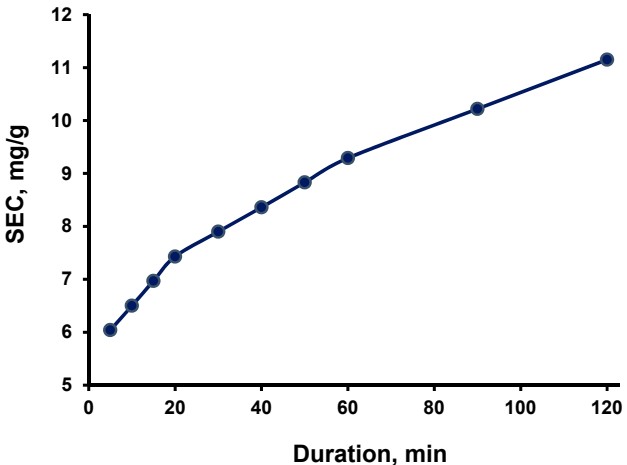

**Figure 10.** Dependence of the SEC value on the contact duration with the AN-2FN anion-exchange resin.

The course of niobium sorption kinetic curves on the Purolite A-100 and AB-17-8 anion-exchange resins (Figures 7 and 8) shows the high kinetic properties of the ion exchangers. It can be seen that an equilibrium comes into the system after 40 min of contact, and the capacity of anion-exchange resins remains practically unchanged for the Purolite A-100 ion exchanger and slightly increases for AB-17-8 with an increase in the process duration.

The good kinetic characteristics of the Purolite A-100 anion-exchange resin can be ensured by the presence of a macroporous structure with a large pore diameter, which

facilitates ion exchange during sorption. The presence of a quaternary trimethylammonium functional group in AB-17-8 resin, which works well not only in acidic but also in alkaline environments (Reaction 21), should provide good adsorption characteristics.

The authors of [32–34] state that AB-17-8 ionite is effective for the sorption of negatively charged complex ions, e.g., molybdenum and tungsten. However, under conditions of niobium sorption, the process is less effective.

The Amberlite IRA 67 anion-exchange resin with high kinetic properties has a low niobium capacity—the SEC does not exceed 7.4 mg/g, which is 7.4 times lower than that of Purolite A-100 (Figure 9).

A number of factors influence sorption. In the Amberlite IRA 67 resin, the ionogenic groups are tertiary amines, which have a weak to medium basicity, which affects the degree of dissociation of the functional groups. However, they are second only to the quaternary ammonium bases in order of basic properties. The resin has a gel structure with a good degree of swelling, which is important to ensure satisfactory permeability of complex niobium anions. Sorption, especially of large anions, is also affected by the amount of cross-linking. The Amberlite IRA 67 resin is based on a cross-linked acrylic structure. All of these factors could influence the niobium sorption values to a greater or lesser extent.

The AN-2FN anion-exchange resin sorbs niobium in insignificant amounts and is characterized by low kinetic properties because equilibrium in the system is not reached even in 120 min (Figure 10). The resin belongs to weakly basic anion exchangers with functional groups in the form of secondary, tertiary aliphatic amino groups and phenolic groups. The resin has a gel structure, and the above factors may also have influenced the niobium sorption performance of the AN-2FN anion-exchange resin.

Thus, as a result of studies, it is shown that the sorption of niobium is more intense and complete when a strong-base Purolite A-100 anion-exchange resin is used.

Further studies were conducted with the promising Purolite A-100 ionite.

The effect of temperature on the sorption rate of niobium ions by the Purolite A-100 anion-exchange resin is shown in Figure 11. As can be seen from Figure 11, the degree of extraction of niobium on the sorbent increases with increasing process temperature from 20 to 50 °C, and the extraction of niobium reaches 45% at a temperature of 50 °C and a process duration of 40 min. When compared with solvent extraction processes [35], where under similar conditions niobium was extracted from a model hydrochloric acid solution using anion-exchange extractants, in particular tertiary amines, the extraction of niobium into the organic phase was 98–99%. The sorption method of niobium extraction, despite the attractiveness of the method, is not well studied. The low extraction rates of niobium indicate the need for additional research.

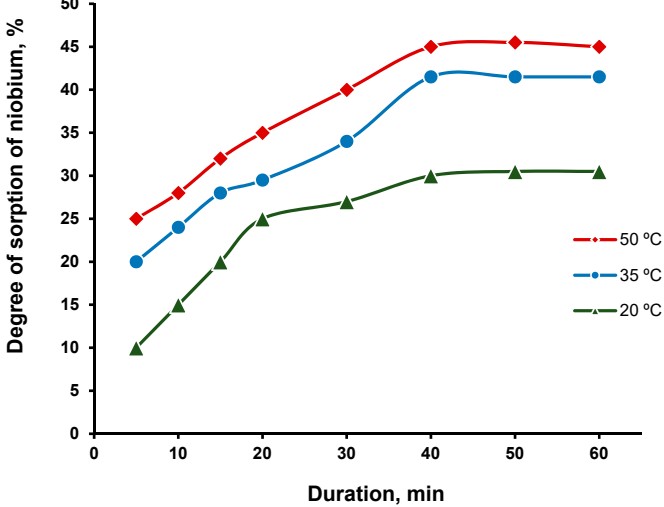

**Figure 11.** Influence of temperature on the niobium sorption degree by Purolite A-100 ion-exchange resin.

Studies of the kinetics of the sorption process can show the limiting stage of the niobium sorption process.

Based on the experimental data, the activation energy of the sorption process was determined using the Arrhenius equation [36]:

$$K = K_o \cdot e^{\frac{-E_{act}}{RT}} \tag{22}$$

where $K_o$ is the pre-exponential multiplier; $E_{act}$ is the activation energy, kJ/mol; R is the universal gas constant, 8.31 J/(mol·K); and T is the absolute temperature, K.

The activation energy value was determined in the temperature range of 293–323 K according to the graphical dependence constructed in coordinates lgK-1/T. According to [37], the diffusion rate constant can be identified with the diffusion coefficient of the limiting stage in calculations of the activation energy carried out by the Arrhenius equation, for kinetic processes whose rate is determined by diffusion.

The kinetic parameters of niobium ion sorption are specified in Table 4. The sorption of niobium from hydrochloric acid solution in the temperature range of 20–50 °C is characterized by the activation energy value equal to 25.32 kJ/mol. It is known [13] that this value of activation energy corresponds to an intermediate region corresponding to the transition from the diffusion to the kinetic regime, i.e., the reasons that inhibit sorption can be both slow diffusion of the ion within the sorbent and the slowing of the chemical reaction itself, which in turn may include either a stage of chemical adsorption or the reaction of ion exchange itself [13].

**Table 4.** Kinetic parameters of niobium sorption with Purolite A-100 anion-exchange resin.

| T, K | $K_d$, mL/g | t, s | $K \cdot 10^{-12}$, $m^2/s$ | $E_{act}$, kJ/mol |
|------|------|------|------|------|
| 293 | 114 | 1800 | 1.731 | |
| 303 | 184 | 1200 | 1.889 | 25.32 |
| 323 | 217 | 900 | 1.902 | |

Thus, the following optimal niobium sorption conditions for the Purolite A-100 anion-exchange resin were established—hydrochloric acid concentration of 5–10 wt.%, process temperature of 35–40 °C and a duration of 40–50 min based on experimental data.

## 4. Conclusions

The study results showed that Purolite A-100 and AB-17-8 anion-exchange resins are characterized by a higher sorption capacity of 54.81 and 28.8 mg/g, respectively, than AN-2 FN and IRA 67. The Purolite A-100 anion-exchange resin exhibited the highest affinity for niobium ions under the conditions studied, with a distribution coefficient of 184 mL/g. Accordingly, the niobium extraction degree in Purolite A-100 was the highest (41.5%), compared with AB-17-8 (27.2%), AN-2FN (7.7%), and Amberlite IRA 67 (5.4%).

It was found that the exchange capacity of Purolite A-100 anion-exchange resin decreases with an increase in hydrochloric acid concentration in the solution. The SEC value was 71.5 mg/g at an acid concentration of 5 wt.%, and the SEC value was ~52–55 mg/g in the interval of 10–15 wt.%.

To study the equilibrium of niobium sorption from the solution on the Purolite A-100 anionite, three well-known isotherm models were applied: Langmuir, Freundlich, and Dubinin–Radushkevich. According to the results of the research, the most acceptable isotherm model is Langmuir, and its correlation coefficient is $R^2 = 0.999$. The obtained data characterize sorption as a process that takes place in a monomolecular layer on the adsorbent that has homogeneous adsorption centers. The obtained value of the average free energy 13.87 kJ/mol indicates the chemical nature of the interaction in the sorption of niobium on the Purolite A-100 anionite.

The optimum niobium sorption conditions for the Purolite A-100 anion-exchange resin were determined based on the experimental data: hydrochloric acid concentration 5–10 wt.%, process temperature 35–40 °C, and duration 40–50 min.

The calculated activation energy value of niobium sorption from hydrochloric acid solution in the temperature range of 20–50 °C was 25.32 kJ/mol, which corresponds to the intermediate region corresponding to the transition from the diffusion to the kinetic mode.

**Author Contributions:** Conceptualization, A.U. and N.L.; methodology, N.L., K.K. and A.M.; software, A.Y. and A.M.; validation, A.U. and N.L.; formal analysis, A.U., Z.K. and N.L.; investigation, N.L., A.Y., K.K. and A.M.; resources, A.Y., K.K. and A.M.; data curation, A.U., N.L. and K.K.; writing—original draft preparation, A.U., N.L. and Z.K.; writing—review and editing, A.U., Z.K. and A.Y.; supervision, A.U.; project administration, A.U.; funding acquisition, A.U. All authors have read and agreed to the published version of the manuscript.

**Funding:** This research is funded by the Science Committee of the Ministry of Science and High Education of the Republic of Kazakhstan, Grant Project No. AP09258788.

**Institutional Review Board Statement:** Not applicable.

**Informed Consent Statement:** Not applicable.

**Data Availability Statement:** The data and results presented in this study are available in the article.

**Conflicts of Interest:** The authors declare that there are no conflicts of interest regarding the publication of this manuscript.

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
