# Peer review of "Studies of Niobium Sorption from Chloride Solutions with the Use of Anion-Exchange Resins"

_processes, doi:10.3390/pr11041288_

Round 1

Reviewer 1 Report

Comments to the authors

Manuscript processes-2311876

Title: Studies of niobium sorption from chloride solutions with the use of anion-exchange resins

General comments:

The manuscript aimed the adsorption of niobium using anion-exchange resins in HCl media. The literature lacks studies regarding application of IX for Nb extraction. Despite that, the manuscript missed scientific comparations and deep discussion.

-       The most common Nb processes involves pyrometallurgy. Please, details the importance of hydrometallurgy in this area.

-       There are studies regarding the use of solvent extraction for Nb separation. Why ion exchange resins would be better than SX?

-       Please, explain why anionic resins are used in acid conditions, since most of the metallic ions are cationic in such condition. According to Eq. 1-3, Nb will be presented as cationic or neutral compounds.

-       The choice of the parameters is not explained.

-       In Results and Discussion section, it is missed the comparison to the literature and in-depth discussion.

-       Adsorption isotherms are missed, and they are crucial to the manuscript.

-       There is no explanation about why Purolite A-100 has the best adsorption results.

-       How the Nb compounds reacts with IX during adsorption?

-       Table 1 – what is the functional group of these resins? Are they the same?

Final decision:

Major revisions.

Author Response

Response to Reviewer 1 Comments

Authors are very much thankful to the reviewer for spending their valuable time for review of our manuscript. We have tried our best to answer and justify all the comments raised by the reviewer and have revised thoroughly as suggested.

Point 1: The most common Nb processes involves pyrometallurgy. Please, details the importance of hydrometallurgy in this area.

Response 1: Due to the fact that the waste from titanium-magnesium and tantalum-niobium production contains low concentrations of niobium as well as significant amounts of other macrocomponents, their pyrometallurgical processing may be inexpedient. Waste recycling is an urgent task for which hydrometallurgical methods can be fully or partially applied. 

The text on page 1-2, lines 40-67 of the edited manuscript has been added for the processing methods and for the importance of hydrometallurgical methods.

Point 2: There are studies regarding the use of solvent extraction for Nb separation. Why ion exchange resins would be better than SX? 

Response 2: Both solvent extraction and ion exchange are new highly selective methods. The advantages of ion exchange over solvent extraction are added in the edited manuscript on page 2, lines 70-75, 78:

“It should be noted that the liquid extraction with organic solvents is an expensive and environmentally harmful method, and in addition to the existing significant advantages, has disadvantages, the main of which are high toxicity, fire hazardous organic extractants. All this imposes higher requirements for safety in storage and operation. In addition to this, with the high cost of organic extractants the question of the possibility of their regeneration is relevant” and

regenerate ion exchange resin...”

Point 3: Please, explain why anionic resins are used in acid conditions, since most of the metallic ions are cationic in such condition. According to Eq. 1-3, Nb will be presented as cationic or neutral compounds.

Response 3: As noted in the manuscript (sentence before equation (1)) “According to [11], [Nb(OH)2Cl4]- ions dominated in 6-8 N hydrochloric acid solutions. Positively charged ions were formed as the concentration of H+ ions increased:”

I.e., positively charged ions, according to [11], are formed at hydrochloric acid concentrations above 8 N HCl. And at HCl concentrations of 6-8 N (20-26%HCl), [Nb(OH)2Cl4]- dominate. Perhaps at lower concentrations, i.e., at 5-20% HCl, negatively charged [Nb(OH)2Cl4]- ions may be present, based on the following assumptions. As can be seen from equations 1 and 2, the equilibrium can shift to the left and right side. Presumably, when the [Nb(OH)2Cl4]- anion interacts with the resin phase from the solution with the transition of the anion to the anionite, the amount of [Nb(OH)2Cl4]- anion in the solution decreases, which should shift the equilibrium towards the formation of the initial substance, i.e., [Nb(OH)2Cl4]- anionite and its further interaction with the ionite phase and the process is supposed to repeat until a certain value of niobium concentration in solution. Therefore, the use of the anionite was of interest.

To explain these assumptions, in the edited manuscript on pp. 4, lines 163-172, the text

«According to equations 1 and 2, depending on the conditions, such as the concentration of the [Nb(OH)2Cl4]- anion, the equilibrium can shift both towards the formation of the [HNbOCl3]+ cation, the neutral complex Nb(OH)2Cl3 or the complex anion [Nb(OH)2Cl4]-. Presumably, when [Nb(OH)2Cl4]- anion from solution interacts with the resin phase and passes to the anionite, the amount of [Nb(OH)2Cl4]- anion in solution decreases, which should shift the equilibrium in the direction of the initial substance formation, i.e. [Nb(OH)2Cl4]- anion and its further interaction with the ion-exchanger phase. The process, presumably, can be repeated up to a certain final value of the niobium concentration in the solution. It is of interest to study the sorption of niobium from chloride solutions using anion exchangers to what extent the process proceeds».  

Point 4: The choice of the parameters is not explained. 

Response 4: Appropriate explanations have been added to the manuscript. Additions in the edited manuscript on pages 6-7, 14 (lines 246-262, 285, 486-488).

Point 5: In Results and Discussion section, it is missed the comparison to the literature and in-depth discussion.  

Response 5: In the "Results and Discussion" section, we have attempted to introduce explanations and discussions of the issues at hand on pp. 12, 13, 14, 15, lines 445-447, 464-465, 486-490, 500-508, 511-513.

Point 6: Adsorption isotherms are missed, and they are crucial to the manuscript.

Response 6: We have added studies on the adsorption isotherms of Langmuir, Freundlich, Dubinin and Radushkevich. In the revised manuscript, a subsection "Study of sorption equilibrium " has been added on pages 7-12.

Point 7: There is no explanation about why Purolite A-100 has the best adsorption results.

Response 7: In the revised manuscript, added explanation of why Purolite A-100 has better adsorption results.

Added text on page 6, lines 246-256

“From the data in Table 1 it follows that Purolite A-100 anionite has a macroporous structure, while the others are of the gel type, which seems to explain its higher niobium capacity compared to the other studied ion exchangers. Ionites with macroporous structure are characterized by the presence of macroporous channels with an ordered structure, which may be more favorable for ion exchange in the sorption of large complex anions of niobium [Nb(OH)2Cl4]- on the sorbent. In this case, for the diffusion of complex niobium ion inside the resin phase provides better availability of counter ions located throughout the volume of the ion exchange resin. The sorption can also be influenced by the hydration of ions. At the same time, it is known that anions with relatively large size are weakly hydrated. The larger the anion and the smaller its charge, it will preferentially pass into the ion-exchange phase [13]”.

Point 8: How the Nb compounds reacts with IX during adsorption?

Response 8: Niobium anions react with IX, as we believe mainly [Nb(OH)2Cl4 ]- .

Reactions (17-21) on pp. 12, lines 428-444.  

Point 9: Table 1 – what is the functional group of these resins? Are they the same?

Response 9: In the "Materials and methods" section we added Table 1 with the characteristics of the ion exchangers. Functional groups of Amberlite IRA-67 and Purolite A-100 anion exchangers are tertiary amino groups; AB-17-8 are quaternary trimethylammonium groups; AH-2FN are secondary, tertiary aliphatic amino groups and phenolic groups.

Reviewer 2 Report

The reviewed manuscript is related to the comparison of four anion exchange resins with respect to sorption of Nb(V). The authors used a hydrochloric solution of niobium as the initial solution, and have shown that Purolite A-100 is the best resin in this case. Despite the importance of this work for further development of the niobium extraction technology, I recommend that this manuscript be revised and resubmitted. Below are my comments:

1. Line 127. The authors mentioned that a solution with 10% HCl was used. Is it wt.%? In section 3.2 the concentration of HCl is varied. Thus, in the experimental section the range of HCl should be mentioned. Thus, the range of HCl content should be mentioned in the experimental section. The same applies to the concentration of Nb (Fig. 2 - change in the concentration of Nb).

2. Please, use the same units in the text (dm3 vs. l; mol/l vs. N etc.).

3. In Fig. 1 and in Table 2 there are several Cyrillic letters.

4. What are functional groups in the used resins? Please, add more explanation of the observed dependencies. 

5. I recommend to use mol/dm3 instead of mg/dm3 in Fig. 2. 

6. Fig. 3. It is clearly seen that a decrease in HCl content leads to an increase in SEC. Why didn't the authors study a lower HCl concentration? Are there any suggestions - will the SEC value increase?

7. Line 228. Should be "equation 1".

8. As far as the lines in all the figures are not the fitted curves or calculated dependencies, it is better to remove the lines.

9. Lines 196-198. Probably, the Nikolsky-Chaos theory is wrong, because many studies reveal the presence of competing ions results in the decrease of sorption. If "The microcomponent amount sorbed by the ion-exchange ion is directly proportional to its concentration in the solution", that means the sorption isotherm has Henry-law dependence. Did the authors try to fit the experimental results with different models of isotherms?

10. Lines 277-278. "E" should be "Eact", "R" is equal to 8.31 kJ/(mol*K), "T" is measured in K.

11. The value of Eact (25.32 kJ/mol) is too precise. What are the uncertainties in all the experiments?

12. There are some typos: line 101 "Cl4"; line 112 "3N" -> "3 N".

Author Response

Response to Reviewer 2 Comments

Authors are very much thankful to the reviewer for spending their valuable time for review of our manuscript. We have tried our best to answer and justify all the comments raised by the reviewer and have revised thoroughly as suggested.

Point 1:. Line 127. The authors mentioned that a solution with 10% HCl was used. Is it wt.%? In section 3.2 the concentration of HCl is varied. Thus, in the experimental section the range of HCl should be mentioned. Thus, the range of HCl content should be mentioned in the experimental section. The same applies to the concentration of Nb (Fig. 2 - change in the concentration of Nb).

Response 1: Acid concentrations are in weight percentages, so throughout the manuscript "wt." is added when referring to weight percentages. The ranges of hydrochloric acid and niobium concentrations studied are given in the edited manuscript on pp. 12 and 7, lines 449 and 282, respectively.

Point 2: Please, use the same units in the text (dm3 vs. l; mol/l vs. N etc.).

Response 2: In the entire manuscript, all units of measurement were reduced to one: dm3 replaced by L, mmol replaced by mg.

Point 3: In Fig. 1 and in Table 2 there are several Cyrillic letters.

Response 3: The Cyrillic letters in Figure 1 and Table 2 have been replaced by Latin letters.

Point 4: What are functional groups in the used resins? Please, add more explanation of the observed dependencies. 

Response 4: Secondary and tertiary amino groups, quaternary trimethylammonium and phenolic groups are present in the resins used. Information on the functional groups and structure of the ion exchangers has been added to the "Materials and Methods" section in Table 1. In the edited manuscript, explanations have been added for results that may depend on the properties of the resins on page 6, lines 246-262.     

Point 5: I recommend to use mol/dm3 instead of mg/dm3 in Fig. 2. 

Response 5: With all due respect, but we had to use mg/L, because in further calculations, mg/L and mg/g were used according to the formulas used. Therefore, mg/l and mmol/g were changed to mg/l and mg/g in the manuscript to conform to a uniform measurement system.

Point 6: Fig. 3. It is clearly seen that a decrease in HCl content leads to an increase in SEC. Why didn't the authors study a lower HCl concentration? Are there any suggestions - will the SEC value increase?

Response 6: HCl concentration below 5 wt.% was not used for the reason that in further studies we plan to leach titanium production wastes containing niobium. It is supposed to extract niobium from the solutions after leaching by sorption. At HCl concentrations below 5 wt.% the leaching will be ineffective. Therefore, we did not anticipate using a concentration below 5 wt% HCl. We also had concerns that at concentrations below 5 wt% HCl, a colloidal precipitate might precipitate due to hydrolysis of the complex anion [Nb(OH)2Cl4]-. In the manuscript, to explain why a lower concentration of HCl was not taken, on pp. 13, lines 464-466, the text «The process was not carried out with concentrations of HCl below 5 wt.% to avoid a colloidal precipitate due to hydrolysis of the complex anion [Nb(OH)2Cl4]- in reaction (3) [11]».

Point 7: Line 228. Should be "equation 1".

Response 7: Yes, that's right. Thank you. Corrected from "equation 3" to "equation 1" (line 463).

Point 8: As far as the lines in all the figures are not the fitted curves or calculated dependencies, it is better to remove the lines.

Response 8: On all figures, the horizontal and vertical grid lines have been removed.

Point 9: Lines 196-198. Probably, the Nikolsky-Chaos theory is wrong, because many studies reveal the presence of competing ions results in the decrease of sorption. If "The microcomponent amount sorbed by the ion-exchange ion is directly proportional to its concentration in the solution", that means the sorption isotherm has Henry-law dependence. Did the authors try to fit the experimental results with different models of isotherms?

Response 9: The hypothesis may well be questionable. However, the excerpt about Nikolsky-Gapon's hypothesis is taken from the literature (reference 17, 18) and, as in reference (18) it is claimed, has been confirmed by a number of researchers. In the manuscript, one reference (18) and part of the text "...which has been confirmed by a number of works,..." is added accordingly, and the sentence "The concentration distribution coefficient of one micro-component is independent of the presence of the other micro-component in solution" is added (lines 272-275 of the edited manuscript).

The results of the studies of Langmuir, Freundlich, Dubinin and Radushkevich isotherm models have been added to the manuscript. Accordingly, an additional subsection "3.2 Study of sorption equilibrium" on pp. 7-12.

Point 10: Lines 277-278. "E" should be "Eact", "R" is equal to 8.31 kJ/(mol*K), "T" is measured in K.

Response 10: In the edited version of the manuscript on lines 531-532, "E" is corrected to "Eact"; "8.31 kJ/mol" to "8.31 J/(mol∙K)", the absolute temperature dimension "K" is added.

Point 11: The value of Eact (25.32 kJ/mol) is too precise. What are the uncertainties in all the experiments?

Response 11: The activation energy value is a calculated value and was rounded to the nearest hundredth. Errors in the calculations are 1-1.5 %.

Point 12: There are some typos: line 101 "Cl4"; line 112 "3N" -> "3 N".

Response 12: Yes, that's right. Thank you. The typos have been corrected on line 141 in the edited version of the manuscript from "Cl4" to "Cl4" and on line 152 from "3N" to "3 N".

Round 2

Reviewer 1 Report

Comments to the authors

Manuscript processes-2311876

Title: Studies of niobium sorption from chloride solutions with the use of anion-exchange resins

General comments:

The manuscript aimed the adsorption of niobium using anion-exchange resins in HCl media. The literature lacks studies regarding application of IX for Nb extraction. The revised version has improved the quality of the manuscript.

-       As observed, the authors evaluated the Dubinin-Radushkevich isotherm, and not Dubinin AND Radushkevich isotherms. Please, correct it in the Abstract.

-       Is there any thermodynamic simulation to state niobium chloride is presented as an anionic compound?

-       Please, compare the results with SX data.

Final decision:

Minor revisions.

Author Response

Response to Reviewer 1 Comments

Authors are very much thankful to the reviewer for spending their valuable time for review of our manuscript. We have tried our best to answer and justify all the comments raised by the reviewer and have revised thoroughly as suggested.

Point 1: As observed, the authors evaluated the Dubinin-Radushkevich isotherm, and not Dubinin AND Radushkevich isotherms. Please, correct it in the Abstract.

Response 1: Corrected in the abstract and throughout the manuscript when mentioning the Dubinin-Radushkevich isotherm.

Point 2: Is there any thermodynamic simulation to state niobium chloride is presented as an anionic compound?

Response 2: According to [11], according to various authors, freshly precipitated niobic acid can be dissolved in concentrated HCl, which produces complex niobium ions whose composition depends on the concentration of H+ and Cl-. In 6-8 N HCl, [Nb(OH)2Cl4 ]- ions dominate. In accordance with this, the formation of niobium anions can be represented by the reaction of interaction of niobic acid with HCl:

Nb2O5∙5H2O + 8HCl = 2[Nb(OH)2Cl4]- + 2H+ + 6H2O

However, the Outokumpu program database that we have (HSC 5.1) lacks the compound and complex niobium ion data necessary for calculations. Therefore, the construction of a thermodynamic model was not yet possible.

Point 3: Please, compare the results with SX data.

Response 3: A comparison of the results obtained with SX is made in the edited version of the manuscript on p. 15, lines 525-533. Accordingly, reference number [35] has been added.

Note: In the edited version of the manuscript, additions or corrections are highlighted in blue.
